# Optimization and Spectrum–Effect Analysis of Ultrasonically Extracted Antioxidant Flavonoids from Persicae Ramulus

**DOI:** 10.3390/molecules29163860

**Published:** 2024-08-15

**Authors:** Qihua Yu, Mingyu Yang, Liyong Yang, Mengyu Li, Ye Yang

**Affiliations:** 1College of Pharmacy, Guizhou University of Traditional Chinese Medicine, Dong Qing Nan Road, Guiyang 550025, China; 18224837415@163.com (Q.Y.); yangmingyu5970@163.com (M.Y.); yangliyong@gzy.edu.cn (L.Y.); 2Guizhou Key Laboratory for Raw Material of Traditional Chinese Medicine, Guizhou University of Traditional Chinese Medicine, Guiyang 550025, China

**Keywords:** persicae ramulus, extraction process, antioxidant, ultra-high-performance liquid chromatography, spectrum–effect relationship

## Abstract

The objectives of this study were to optimize the ultrasonic-assisted flavonoid extraction process from PR and to establish fingerprints in order to analyze the spectrum–effect relationship of antioxidant activity. The ultrasonic-assisted flavonoid extraction process from PR was optimized using RSM, and the fingerprints of twenty-eight batches of flavonoids from PR were established using UHPLC. Meanwhile, the in vitro antioxidant activity of PR was evaluated in DPPH and ABTS free radical-scavenging experiments. Then, the peaks of the effective antioxidant components were screened using the spectrum–effect relationships. The results show that the optimal extraction yield of flavonoids from PR was 3.24 ± 0.01 mg/g when using 53% ethanol, a 1:26 (g/mL) solid–liquid ratio, and 60 min of ultrasonic extraction. Additionally, the clearance of two antioxidant indices by the flavonoids extracted from PR had different degrees of correlation and showed concentration dependence. Simultaneously, the similarity of the UHPLC fingerprints of twenty-eight batches of PR samples ranged from 0.801 to 0.949, and four characteristic peaks, namely peaks 4, 12, 21, and 24, were screened as the peaks of the components responsible for the antioxidant effect of PR using a GRA, a Pearson correlation analysis, and a PLS-DA. In this study, characteristic peaks of the antioxidant effects of PR were screened in an investigation of the spectrum–effect relationship to provide a scientific basis for the study of pharmacodynamic substances and the elucidation of the mechanism of action of the antioxidant effect of PR.

## 1. Introduction

Persicae ramulus (PR) is the dried branch of the peach *Prunus persica* (L.) Batsch, which belongs to the Rosaceae family. It can promote circulation, remove blood stasis, and moisten the intestines to help pass stools, and it is used to treat dysmenorrhea caused by blood stagnation, abdominal pain caused by postpartum stasis, traumatic injuries, and intestinal carbuncle [1,2]. A chemical study showed that PR contains flavonoids, alkaloids, glycosides, phenylpropanoids, organic acids, and volatile oils [3]. Among them, flavonoids have good pharmacological research value and development and utilization potential [4,5]. However, studies on PR have mainly focused on the qualitative analysis of its chemical composition and volatile oils [3], and research on the optimization of the extraction technique of flavonoids is currently lacking.

Ultrasound-assisted extraction (UAE) is a method that uses the acoustic cavitation effect produced by ultrasonic waves to promote the fast dissolution and precipitation of components [6,7]. Compared with the traditional extraction process, the ultrasonic extraction method has the advantages of being quick, highly efficient, green, and economical [8]. However, inappropriate ultrasound extraction conditions can have unfavorable effects on the extraction rate. The Box–Behnken design (BBD) in response surface methodology (RSM) is a data model-fitting method that can reduce the number of experiments and identify the factor that has a significant effect on the test via mathematical modeling and integrating experimental design [9,10]. To date, RSM has been comprehensively applied and gradually improved for the modeling and optimization of ultrasound-assisted extraction of flavonoids. Thus, RSM was applied to optimize the ultrasonic flavonoid extraction process from PR in this study.

Free radicals are intermediate products of the body’s metabolism, depending on the antioxidant system used to maintain a particular body homeostasis process [11]. When free radicals excessively accumulate in the body, oxidative stress occurs and leads to a chain reaction, causing oxidative damage to cells and tissues and leading to the occurrence of a variety of diseases, such as neurodegenerative diseases, tumors, and cancer [12,13]. Studies have shown that antioxidants can effectively delay the oxidation of substances and neutralize free radicals, thereby reducing the pathological process mediated by oxidative stress [14]. Therefore, coadjutant treatment for diseases caused by free radical accumulation can be completed with a moderate antioxidant intake. Synthetic antioxidants can lead to endocrine disruption and carcinogenic lesions [15]; thus, the search for green and effective natural antioxidant active components has become one of the current research hot spots.

Flavonoids are a class of compounds with a C6-C3-C6 structure as a basic skeleton, and they have various biological and pharmacological activities, including antioxidant, anti-inflammatory, immunomodulatory, cardioprotective, and anticancer activities [16,17,18]. Currently, flavonoids are widely employed in the pharmaceutical and health food fields, with broad research prospects. As natural antioxidants, they can effectively scavenge unnecessary free radicals in the body and maintain the organism’s free radical homeostasis balance, thereby inhibiting the occurrence of oxidative stress reactions [19]. However, it remains uncertain whether flavonoids in PR have pharmacological antioxidant effects.

The fingerprinting technique is a comprehensive and quantifiable identification method [20] which mainly focuses on determining the content of the chemical constituents of medicinal materials rather than screening their pharmacodynamic components directly [21]. The spectrum–effect relationship is an effective method for screening bioactive compounds and verifying the pharmacodynamic material basis of medicinal materials, and it is determined by combining fingerprinting technology with pharmacodynamics [22]. An analysis of the spectrum–effect relationship can address the above issues and lay a foundation for the pharmacology research of medicinal materials. In the study of spectrum–effect relationships, mathematical models provide new ideas for screening pharmacodynamic components, but each model has its own limitations. A Pearson correlation analysis can evaluate the degrees of the positive and negative correlations of different variables, but it cannot assess the overall effect of multiple components [23]. A gray correlation analysis can evaluate the contribution of components to efficacy [24], but it cannot describe the direction of the degree of correlation of each component with efficacy. Therefore, a combined PLS-DA analysis based on these methods can further clarify the active components that are positively and negatively correlated with efficacy.

At present, there are few studies on the dose–effect and spectrum–effect relationships between the flavonoid and antioxidant activity of PR, which limits the in-depth study of PR’s pharmacodynamic material basis and clinical efficacy. Therefore, in this study, the ultrasound-assisted process of flavonoid extraction from PR was optimized using RSM, and the free radical-scavenging experimentation methods of 2,2-diphenyl-1-picrylhydrazyl (DPPH) and 2,2′-diazide-bis (3-ethylbenzothiazoline-6-sulfonic acid) diammonium salt (ABTS) were used to evaluate the antioxidant activity of PR. Simultaneously, an ultra-high-performance liquid chromatography (UHPLC) system was used to establish the chromatographic fingerprints of twenty-eight batches of flavonoids from PR, and the spectrum–effect relationship was used to screen the ingredient peaks of antioxidant activity using a gray relational analysis (GRA), Pearson correlation analysis, and partial least-squares discriminant analysis (PLS-DA), further providing a new horizon for further studies on the pharmacological effects of PR.

## 2. Results

### 2.1. Methodological Investigation

#### Content Determination Results

The RSD values for the precision, repeatability, stability, and recovery experiments for content determination were 0.13%, 2.18%, 0.57%, and 2.93%, respectively, and they were all less than 3.0%, as shown in Appendix A. It was found that the precision of the UV–vis spectrophotometer and the repeatability and accuracy of the method were good, and the chemical components in the PR sample were stable for 12 h. The instrument and method selected can be used to determine the flavonoid contents of PR.

### 2.2. Optimization of Single-Factor Tests

After a preliminary investigation of the two extraction methods of ultrasonic extraction and heating reflux with water, in which methanol and ethanol were used as extraction solvents, it was found that the extraction rate of flavonoids from PR was significantly higher when ethanol was selected as the solvent for ultrasonic extraction. This indicates that selecting ethanol as an ultrasonic extraction solvent is an effective method for extracting flavonoids from PR. Further, in subsequent experiments, the ultrasonic flavonoid extraction process from PR was optimized, and the test results are shown in Figure 1 and Appendix A.

#### 2.2.1. Extraction Time

The effect of extraction time on the yield of flavonoids from PR was investigated under the conditions of a 1:25 (g/mL) solid–liquid ratio and 50% ethanol volume fraction. The yield of flavonoids presented a trend of first increasing and then decreasing over 50 min, steadily improving and reaching a maximum at 50 min, and then leveling off after 50 min with the increase in the extraction time, as shown in Figure 1a. This was possibly due to some compounds in the target extract being sensitive to the acoustic cavitation reaction caused by the ultrasound. The consequent degradation resulted in the compound structure being destroyed and in the extraction rate decreasing [25]. Therefore, an extraction time of 50 min was deemed appropriate through experimental screening.

#### 2.2.2. Solid–Liquid Ratio

Figure 1b shows that the yield of flavonoids increased steadily and reached the highest value with an increasing solid–liquid ratio. When the solid–liquid ratio was higher than 1:25 (g/mL), the extraction rate showed a trend of beginning to consistently reduce. The yield of the target extracts may have subsequently decreased when the medicinal materials could not be completely infiltrated. Because the solid–liquid ratio was so low, the contact area was limited. The active ingredient from the sample was significantly dissolved when the applied amount of solution reached an appropriate volume, which may have been due to the increased contact area between the material and the solvent. The content of target extracts was reduced because the excessive addition of solution diluted the sample [26]. After comprehensive consideration, 1:25 (g/mL) was selected as a suitable solid–liquid ratio.

#### 2.2.3. Ethanol Volume Fraction

The effect of the ethanol volume fraction on the yield of flavonoids from PR can be observed in Figure 1c, with an extraction time of 50 min and a solid–liquid ratio of 1:25 (g/mL). The yield of flavonoids gradually improved with an increasing ethanol concentration in the range of 40% to 50%. Then, the yield significantly decreased when the ethanol concentration exceeded 50% to 80%. Flavonoids usually exist in plants in the form of glycosides, while a small portion exists in a free form as aglycones. The water solubility of flavone glycosides is strong; thus, their yield is controlled when the ethanol volume fraction is steadily increased. Free aglycones are easily dissolved in organic solvents, so a constant increase in ethanol concentration is more conducive to promoting the dissolution of free flavonoids. However, other components will be extracted at the same time, thus leading to a decrease in flavonoid content [27]. Consequently, a 50% ethanol volume fraction was deemed to be suitable for screening.

### 2.3. Optimization Test Results of BBD-RSM

The extraction time, solid–liquid ratio, and ethanol volume fraction were selected as effect factors based on the single-factor test results, and the yield of flavonoids from PR was used as the evaluation index [28]. Thus, response surface experiments consisting of three factors and three yield levels of flavonoids from PR were conducted, and the results are shown in Appendix A. These experiments were carried out in triplicate.

#### 2.3.1. Analysis of Variance

The experimental results were fitted and analyzed using Design Expert 13.0 software to explore the effects and determine the optimal values of the three independent variables on the yield of flavonoids from PR. The following regression equation was obtained:Y = −24.73 + 0.25A + 1.35B + 0.16C − 0.01AB + 0.01AC + 0.01BC − 0.01 A^2^ − 0.02B^2^ − 0.01C^2^.

In Figure 2 and Appendix A, the results show that the F-value was 35.74 (*p* < 0.01), indicating that the model was very significant, and the *p*-value of the lack-of-fit term was 0.07, which was higher than 0.05 and not significant. The R^2^ value was 0.98, and the prediction value of the model was close to the actual value, indicating that the model is appropriate for the optimization of the flavonoid extraction process from PR. The size ordering of the effect of the three independent variables on the extraction of flavonoids was as follows: solid–liquid ratio > ethanol volume fraction > extraction time.

The RSM results of the flavonoid yield from PR are shown in Figure 2. A larger radian of the response surface and isocontours approaching ovals indicate that the interaction between the two factors is significantly greater [29]. As shown in Figure 2, the ordering of the response surface radian of the interaction between any two factors was 2e > 2a > 2c, and the ordering of the isocontour graphs presenting an oval degree was 2f > 2b > 2d. In the response surface plots of the interactions between any two factors of A, B, and C, it was observed that the response surface radian of 2c was the steepest, indicating that the interaction between B and C had the greatest impact on the yield of flavonoids from PR. When observing 2a and 3c, it was found that the radian of the side of the solid–liquid ratio was steeper, and the surface plot of the extraction time was the gentlest compared with that of the other two factors. This shows that the solid–liquid ratio had an extremely significant effect and that the effect of the extraction time on the response values was small. In the variance analysis results, the F-value of B was the largest compared with that of the other two factors, and its effect was significant, with a greater F-value and a lower *p*-value; this shows that the effect of factor B was the most pronounced, followed by the effect of factor C, with the effect of factor A being the lowest. This indicates that the response surface analysis results were consistent with the analysis of variance results in terms of the significance of the interactions of A, B, C, BC, AB, and AC.

#### 2.3.2. Verification Experiment

The optimum extraction conditions determined from the RSM analysis results were as follows: an extraction time of 48.79 min, a solid–liquid ratio of 1:26.12 (g/mL), and an ethanol volume fraction of 53.27%. Under these conditions, the theoretical yield of flavonoids was 3.35 mg/g. The extraction factors were adjusted based on the feasibility of practical conditions, resulting in an extraction time of 50 min, a solid–liquid ratio of 1:26 (g/mL), and an ethanol volume fraction of 53%. The PR sample was accurately weighed to further confirm the reliability of the response model. From the three parallel experiments, the average flavonoid yield was found to be 3.24 ± 0.01 mg/g, which had a relative deviation of 3.73% compared with the predicted value. The experimental result was similar to the predicted value; this proves that the extraction conditions obtained using the Box–Behnken design response surface method were valid and reliable, and it demonstrates that adopting the above process to extract flavonoids from PR is a stable and feasible approach.

#### 2.3.3. Determination of Flavonoid Content in Different Batches of PR

The flavonoid content in twenty-eight batches of PR was determined using the optimal extraction process after response surface optimization, and the results are shown in Figure 3 and Appendix A. The batch with the highest yield of flavonoids was S1, at 3.76 ± 0.01 mg/g, and the batch with the lowest yield was S21, at 0.74 ± 0.01 mg/g. The results in the table were converted to a plot using GraphPad software (version 9.5.0).

#### 2.3.4. HCA of Flavonoid Content in PR from Different Areas

SPSS software (version 26.0) was used to analyze the results in Section 2.3.3 and conduct a cluster analysis for the determination of flavonoid yield; a dendrogram of the results is shown in Figure 4. Twenty-eight batches of PR samples were grouped into five major clusters by adopting the clustering method of intergroup join, with a Euclidean distance of d = 10. In the content determination results, samples with a flavonoid yield higher than 3.31 ± 0.35 mg/g were clustered into the first category, samples with a flavonoid content ranging from 2.63 ± 0.04 mg/g to 3.24 ± 0.01 mg/g were clustered into the second category, samples with a flavonoid content ranging from 1.33 ± 0.06 mg/g to 1.64 ± 0.03 mg/g was clustered into the third major category, and samples with a flavonoid content lower than 0.93 ± 0.12 mg/g were clustered into the fourth major category; the remaining batches were clustered into the fifth category. Combining the HCA results with the origin of the PR batches revealed differences in the flavonoid content in batches with a common origin. The factors that affect the yield of flavonoids from PR may be the climate, temperature, and rainfall of each place of collection and the collection time.

### 2.4. Antioxidant Activity

Various concentrations (0.03–0.15 mg/mL) of PR flavonoid extracts were used to conduct antioxidant studies, and the antioxidant activities of twenty-eight batches of PR samples were determined using the DPPH and ABTS radical-scavenging methods. The results show that the DPPH and ABTS clearance rates increased linearly with the concentration.

SPSS 26.0 software was used to analyze the data from the antioxidant activity assays, which are shown in Appendix A and transformed in Figure 5. In the DPPH free radical-scavenging assay, the S4 sample collected from Dangwu showed the highest scavenging activity, which reached 87.89 ± 0.63%, and the S16 sample was the batch with the lowest scavenging rate, which was 40.19 ± 0.37%. In the ABTS free radical-scavenging assay, the S21 sample was the strongest batch, with a clearance rate of 87.56 ± 0.40%, and the S16 sample was the weakest batch, with a clearance rate of 24.73 ± 0.49%, which shows that using only one indicator to evaluate antioxidant activity has limitations. A preliminary study of the antioxidant capacity of the flavonoid in PR revealed that the clearance rates of the DPPH and ABTS free radicals by the flavonoid extracts from the PR batches of S1, S5, S7, S9–S17, S19, and S22 were consistent with the flavonoid yield, which indicates that the levels of antioxidant activity differed with the flavonoid content in the extracts.

Significant differences were found in the antioxidant activities of the twenty-eight batches of samples; thus, using two approaches for a comprehensive evaluation can verify whether flavonoid extracts have antioxidant activity well.

### 2.5. Establishment of UHPLC Fingerprinting

#### 2.5.1. System Suitability Investigation Result

Through the experimental optimization of the system suitability conditions, the optimal conditions were determined for the UHPLC fingerprinting of the twenty-eight batches of PR samples. Regarding the elution conditions, the mobile phase gradient in Section 4.8.1 was used, as shown in Appendix A, and the other chromatographic conditions were as follows: a detection wavelength of 290 nm, an analysis time of 30 min, 0.01% formic acid aqueous solution (A) and 0.01% formic acid acetonitrile (B) as the mobile phase, a flow rate of 0.2 mL/min, and a column temperature of 25 °C. Under these conditions, the UHPLC fingerprinting had more chromatographic peaks, and the peak intensities, peak shape, chromatographic resolution, and stability were better. Additionally, the baseline was steady, and the characteristic peaks in the samples completely appeared within 30 min. The results are shown in Appendix A.

#### 2.5.2. Methodology Validation of UHPLC

In order to obtain stable and reproducible UHPLC fingerprinting, the test solution was determined according to the chromatographic conditions in Section 4.8.1, and a chromatogram was recorded to validate the UHPLC fingerprinting methodology. The results revealed that the blank solvent did not interfere with the determination. The PRA values of 35 common peaks in the precision, reproducibility, and stability tests were 1.95%, 1.98%, and 1.96%, respectively, and the RRT values of these 35 common peaks in the same experiments were 2.72%, 2.62%, and 1.79%, respectively. The RSDs of the RRT and RPA of the 35 common peaks were less than 3.0% (as shown in Appendix A), which indicates that the precision of the instrument used in the trial was good and that the components in the above samples remained stable over 12 h. Thus, the method was deemed suitable for PR UHPLC fingerprinting with good precision, reproducibility, and stability.

#### 2.5.3. Establishment of the UHPLC Fingerprinting of PR Samples and Identification of Common Peaks

Chlorogenic acid, neochlorogenic acid, caffeic acid, quercetin standard solution, and test solutions of the twenty-eight batches of PR were determined according to the chromatographic conditions in Section 4.8.1. The chromatograms of the sample solutions were imported into CFSESTCM software (version 2012), and the median method was used to generate the control map R. In addition, the chromatogram of S1 was used as the reference, and the widths of the temporal windows were set to 0.1 min. Then, the Mark peaks were matched with multipoint correction to generate a UHPLC fingerprinting overlay of the PR samples (Figure 6), and chromatograms of the 35 common peaks and control solution (Figure 7) were obtained. Among them, four peaks were identified as neochlorogenic acid (peak 5), chlorogenic acid (peak 6), caffeic acid (peak 8), and quercetin (peak 26). Among the 35 common peaks, peak 17 had a suitable retention time, peak area, and good separation, and it was observed to be steady in the chromatograms of all samples. Therefore, peak 17 was assigned as the reference to calculate the relative retention times (RRTs) and relative peak areas (RPAs) (as shown in Appendix A) of each characteristic peak.

#### 2.5.4. Similarity Analysis (SA)

The similarities between all chromatograms of the twenty-eight batches of PR and the reference chromatographic profile were analyzed to verify the quality differences between all samples using the professional software SESCFTCM (version 2012), and the similarity was then calculated (Table 1). By comparing each PR sample’s chromatogram with the fingerprinting reference (R), it was found that the similarity values were in the range of 0.801–0.949, and these values were uniformly larger than 0.8. The similarity data indicate that the twenty-eight batches of PR samples had a certain high similarity and were relatively stable and that the chemical compositions were consistent but different.

#### 2.5.5. Hierarchical Cluster Analysis (HCA) of UHPLC

The areas of the 35 common peaks of the twenty-eight batches of PR were input into SPSS26.0 software. The interval of intergroup connection and Euclidean distance methods were adopted to analyze the HCA results of the PR samples, and Origin software (version 2021) was used to establish a visual analysis heatmap, as shown in Figure 8. The HCA results show that the PR samples were mainly divided into three categories when the connection distance between the groups was d = 15. The S1–S6 and S8–S27 samples were clustered into one category, the S7 sample was clustered into one category, and the S28 sample was clustered into another category. A comparison with the HCA results of the flavonoid content in PR revealed that these samples had good similarity but that there were differences in the chemical composition of the contents, which proves that the samples were not collected at the same time.

#### 2.5.6. Principal Component Analysis (PCA)

To verify the composition differences in each batch of samples, the data on the area of the 35 common peaks in the chromatograms were used as variables and imported into SPSS26.0 software for Z-score standardization. Then, the data were subjected to a principal component analysis (PCA). The PCA results showed that a total of nine principal components (PCs) were generated with PC extraction using a standard eigenvalue of more than 1, and the cumulative variance contribution rate reached 86.96%, which was more than 85%, indicating that the nine PCs could represent most of the fingerprinting information of the 35 common peaks of the PR samples. As shown in Appendix A, the variance contribution rates of the principal components were calculated via the factor extract and factor rotation results.

Based on the score coefficients in Appendix A, the formula for the principal component score factor (PCSF) of the nine principal components is Z1 = −0.069X1 + 0.02X2 + 0.168X3 + 0.061X4 − 0.011X5 + … + 0.079X35, where X1–X35 and Z represent the standardization data and the PCSF, respectively. Then, the same method was used for the calculation of Z2 to Z9, and the calculation result of the score factor was applied to calculate the principal component score (PCS, y) at the same time. The comprehensive scores (Y) of the PR samples were calculated by using the PCS and variance contribution rate results of the PCs as weights to perform linear weighting, as shown in Appendix A.

To further visualize the results, the data were imported into Simca software (14.0 version) to construct a principal component analysis score plot, as shown in Figure 9. The samples were mainly clustered into different areas. S28, which was distributed outside of the circle, was significantly different from the other samples in the discrete degree of the PCA score plot. The PCA results were consistent with the HCA and SA results, and they indicated differences in the chemical composition of the samples, which proves that the samples were not collected at the same time.

### 2.6. Studies on the Spectrum–Effect Relationship Analysis

#### 2.6.1. Grey Relational Analysis

As shown in Table 2, the relationship between the 35 common peaks and the DPPH and ABTS radical-scavenging activities were examined using a GRA. In the ABTS assay, peaks with a correlation from high to low were ranked in the following order: P1  >  P12  >  P13  >  P28 >  P19  >  P2  >  P10 >  P20  >  P9  >  11  >  P26  > P3  >  P22  >  P16  >  P18  >  P4  >  P8  >  P34  >  P21  > P14  >  P24  >  P5  >  P25  >  P35  >  P15  >  P33  >  P17  > P27  >  P23  >  P29  >  P30  >  P31  >  P7  >  P32  >  P6 for DPPH assay, P1 > P3 > P27 > P21 > P11 > P22 > P14 > P12 > P9 > P33 > P13 > P26 > P4 > P5 > P34 > P32 > P28 > P24 > P19 > P15 > P20 > P35 > P16 > P18 > P7 > P6 > P17 > P8 > P25 > P10 > P23 > P31 > P30 > P29 > P2. The gray relational grade was greater, and the correlation between the common peaks and the efficiency was stronger. Therefore, the results show that all common peaks made a significant contribution to the antioxidant activity of DPPH and ABTS, with each correlation degree being more than 0.9, which indicates that the antioxidant activity of the PR samples was the result of the synergy of multiple components.

#### 2.6.2. Pearson Correlation Analysis

In this study, the common peak area from the UHPLC fingerprints of the PR flavonoid extracts and the scavenging rate data after Z-score normalization were imported into Origin software (version 2021), and the relationship was evaluated using a Pearson correlation analysis. The results (Figure 10) show that the areas of peaks 1–6, 10–13, 15, 16, 19–21, 25–28, 31, and 33–35 were positively correlated with the DPPH- and ABTS-scavenging activity values. It could be hypothesized that 23 shared peaks correspond to components that may be related to the antioxidant activity of the PR flavonoid extracts, and the larger the peak area, the greater the free radical-scavenging rate of the PR flavonoid extracts and the stronger the antioxidant activity.

#### 2.6.3. Partial Least-Squares Discriminant Analysis (PLS-DA)

The DPPH and ABTS free radical-scavenging rates of the PR samples were used as dependent variables (X), and the common peak areas were used as the independent variables (Y). A PLS-DA was performed using SPSS 26.0 software, with the latent factor number set to 5. Then, the standardized regression coefficients and the VIP values were calculated to evaluate the correlation between PR and antioxidant activity. The results are shown in Appendix A, and the data of the peak area and the antioxidant test were transferred into Simca14.0 software to construct a VIP value plot (Figure 11) and normalized regression coefficient plots (Figure 12).

The VIP value is known to be an influencing variable that explains the importance of the independent variable (Y) to the dependent variable (X). The larger the VIP value, the stronger its explaining ability and, thus, the greater the contribution of the corresponding chromatographic peaks to the pharmacodynamic activity. The VIP values between the common peak areas of all the PR samples and the DPPH and ABTS free radical-scavenging ability values were positive, as shown in Appendix A, which indicates that the components represented by the chromatographic peaks were positively associated with the antioxidant effects. Additionally, VIP > 1 indicates that the independent variable makes a significant contribution to the dependent variable. By combining the two methods of evaluation, it was found that the components represented by peaks 4, 6, 8, 11–14, 17–18, 21–22, 24, 26, and 27 made significant contributions to the antioxidant effect in the PR samples.

In the PLS-DA statistical analysis, a positive correlation was indicated between the constituent corresponding to the chromatographic peak and pharmacodynamic activity when the standardized partial regression coefficient was positive, and vice versa. As shown in Appendix A and Figure 11, the standardized regression coefficients of chromatographic peaks 3, 4, 9, 10, 12, 15, 19–21, 24, 26, 29, 31, and 33 were positive, which indicates that the constituents corresponding to these chromatographic peaks may be the main constituents accounting for the antioxidant effects of PR.

In conclusion, it was found that the characteristic peaks obtained by using two indicators in a common evaluation method were basically consistent. Among them, the order of chromatographic peaks 4, 12, 17, 21, 24, and 26 was different; however, the VIP values were all greater than 1, and the standard regression coefficients were all positive values, which indicates that the chemical components represented by the abovementioned chromatographic peaks may have made a greater contribution to the antioxidant effect of PR.

## 3. Discussion

### 3.1. Optimization of Extraction Process and Antioxidant Activity

Oxidative stress, an imbalance in the oxidative system that can lead to cellular or tissue damage, underlies the development of numerous diseases [30]. Recent pharmacological studies have shown that flavonoids have potent biological properties, including antioxidant activity [31]. The antioxidant activity of flavonoids in PR has not been investigated; therefore, in this study, using a one-way test to select the optimum ranges of three factors, the effects of the extraction solvent (40%, 50%, and 60% ethanol), extraction time (40, 50, and 60 min), and solid–liquid ratio (1:20, 1:25, and 1:30) on the extraction effect of PR test solutions were investigated using a Box–Behnken response surface test. The flavonoid yield of twenty-eight batches of PR was determined to range from 0.74 ± 0.01 mg/g to 3.76 ± 0.11 mg/g. This indicates that there were differences in flavonoid yield between the PR batches, which may be related to the harvesting season, storage environment, and other factors. Furthermore, a novelty of this study is that it is the first to investigate the antioxidant effect of flavonoids in PR. A preliminary study of the antioxidant capacity of PR revealed that its scavenging rates for two types of free radicals varied with flavonoid yield, which indicates that the flavonoid extract from PR has antioxidant capabilities. However, the extraction in this experiment was only crude, the polarity of the solvents used was high, and other active components possessing antioxidant effects were present in the extract in addition to flavonoids. Furthermore, the gelatinous components of PR or the dissolution of other impurities may have interfered with the experiment. Therefore, it was also necessary to isolate and purify the flavonoid component of PR and further verify the in vivo antioxidant activity of this component.

### 3.2. Establishment of Fingerprinting and Relevant Stoichiometric Analysis

Currently, the “Similarity Evaluation System for Chromatographic Fingerprinting of Chinese Medicines (2012 version)” is used to calculate similarities. However, this method cannot determine the content of chemical constituents, and it cannot reflect changes in the content of Chinese medicines [32]. Therefore, a high degree of similarity only indicates that different batches of samples have similar constituents and does not indicate the component contents in each batch of samples [33]. In this study, the following chromatographic conditions were determined: a wavelength of 290 nm, an analysis time of 30 min, a mobile phase consisting of 0.01% formic acid water (A) and 0.01% formic acid acetonitrile (B), a flow rate of 0.2 mL/min, and a column temperature of 25 °C. The typical UHPLC fingerprinting of twenty-eight batches of PR was established, and it showed that the chemical compositions of PR with different origins were basically consistent but that the contents were slightly different. In the fingerprinting similarity evaluation, the fingerprinting chromatographic similarity values of PR samples from different areas had high chemical composition similarity. Among them, samples S20 (0.801) and S21 (0.949) originated from the same place and had largely different similarity values, and the similarity between S2 and S28, which were from different places of origin, was also low. The differences in chemical component content may be the cause of this low similarity. In the HCA results, the clustering between the PR samples of different origins appeared to cross. By combining the common peak areas and PCA results, it was found that samples S2 and S20 had relatively large differences to the other samples in terms of the peak areas in the fingerprints. This may be the main reason that caused the composite scores of S2 and S20 to be different from those of the other samples. It was found that the PCA results had a certain correlation with the peak area of PR, and the chemical composition information of PR will be analyzed in depth with this classification basis. The SA, HCA, and PCA results consistently showed that the fingerprinting of the different PR samples had similar peaks, but the extracts from and the chemical composition content in the PR samples significantly varied. The reasons for the variability in the chemical composition content between each PR batch may be related to factors such as the harvesting season, the geographical environment, and the storage environment, and it is still necessary to further investigate the above influencing factors at a later stage in order to provide a reference basis for improving the efficiency of extracting this type of component.

### 3.3. Spectrum–Effect Relationship Analysis

The study of the spectrum–effect relationship is a key method to clarify the pharmacodynamic material basis of traditional Chinese medicines [34]. Combining chemometric methods to analyze the data of characteristic fingerprinting profiles and pharmacodynamics can effectively realize the quality control of the chemical composition of medicinal materials [35]. A GRA applies gray correlation to reflect the strength, magnitude, and order of the relationships between multiple factors and variables, and a Pearson correlation analysis is able to identify common peaks that are strongly correlated with pharmacological activity. However, these two methods cannot determine whether the pharmacological effect is promotion or inhibition [36]. PLS-DA is a statistical method used to build a linear regression mathematical model of independent and dependent variables, combining a PCA, a canonical correlation analysis (CCA), and a multiple linear regression analysis, and it can directly reflect the relationship between the independent and dependent variables as a positive or negative correlation [37]. When VIP > 1, it indicates that the screened component peaks play an important role in the examined pharmacological activity. Therefore, the spectrum–effect relationship of the antioxidant capacity of the twenty-eight batches of PR was analyzed by using the above three methods as a chemical evaluation method. In the GRA, the correlation between 35 common peaks and antioxidant activities was greater than 0.9. In the Pearson correlation analysis, peaks 1–6, 10–13, 15, 16, 19–21, 25–28, 31, and 33–35, which had positive correlation coefficients, were screened as the antioxidant component peaks. In the PLS-DA, the VIP values of peaks 4, 6, 8, 11~14, 17, 18, 21, 22, 24, 26, and 27 were all greater than 1, indicating that these peaks had a more significant effect on the antioxidant activity in PR, and the regression coefficients of peaks 3, 4, 9, 10, 12, 15, 16, 19~21, 24, 29, 31, and 33 were positive. By combining the above three statistical models with a gray correlation of >0.9, positive correlation coefficients, VIP > 1, and positive regression coefficients as the screening conditions, the four peaks 4, 12, 21, and 24 were found to be the main PR components exerting antioxidant activity, which indicates that the antioxidant activity of PR may be the consequence of the synergistic effect of several components and might not be dominated by a single substance. In addition, the results of the areas of the common peaks indicated that the areas of peaks 17, 18, 19, 22, 23, and 29 were larger than those of the other peaks of the PR samples. As such, the above chromatographic peaks should also indicate the components responsible for the antioxidant activity of PR, and further in-depth studies are required in the future. In this study, only some of the peaks, namely neochlorogenic acid (peak 5), chlorogenic acid (peak 6), caffeic acid (peak 8), and quercetin (peak 26), were identified, and the other chromatographic peaks were not successfully identified. We hope to analyze the structure of each chemical constituent using mass spectrometry and other high-specificity techniques, combined with a pharmacodynamic analysis, to further identify the active antioxidant substances.

## 4. Materials and Methods

### 4.1. Reagents and Materials

2,2-diphenyl-1-picrylhydrazyl (DPPH; AFBJ3115), 2-azino-bis (3-ethylbenzothiazoline-6-sulfonic acid) (ABTS; AF21091251), and reference standards of neochlorogenic acid, caffeic acid, chlorogenic acid, quercetin, and rutin (purity ≥ 98%) were purchased from Chengdu Alfa Biotechnology Co., Ltd. (Chengdu, China). Formic acid (K2215763) was obtained from Aladdin Bio-Chem Technology Co., Ltd. (Shanghai, China). Chromatographic-grade methanol and acetonitrile were provided by TEDIA (Fairfield, OH, USA). All other chemical reagents were of analytical grade.

Twenty-eight batches of PR were collected from different natural growing areas. Sample origins and batch numbers are shown in Appendix A. These samples were identified by associate professor Ye Yang at the Guizhou University of Chinese Medicine, in accordance with the 2020 edition of the *Chinese Pharmacopoeia*.

### 4.2. Preparation of Working Solutions

Regarding the ultrasound-assisted extraction conditions (with an ultrasound power of 60% and an ultrasound frequency of 40 kHz), 1 g of PR powder was precisely weighed, 25 mL of 53% ethanol aqueous solution was added, and ultrasonic extraction was carried out for 50 min at 60 °C. The sample extract solutions were filtered to obtain the supernatant for the determination of the flavonoid content in PR and for the DPPH and ABTS antioxidant activity assays.

Referring to the extraction method of Wang et al. [38], rutin was accurately weighed and dissolved in 50% ethanol to prepare standard solutions of 0.40 mg/mL.

Different volumes of the standard solutions were accurately transferred to 10 mL volumetric flasks to construct a standard curve, plotting the concentration as the abscissa and the absorbance as the ordinate, as shown in Appendix A. This revealed that the concentration of the rutin standard solution was well within the range from 0.06 to 0.4 mg/mL.

Referencing the method in [39], 1.25 g of aluminum chloride salt solid was accurately weighed to prepare a 5% aluminum chloride solution, and 2.31 g of acetic acid was accurately weighed to prepare 0.2 mol/L of an acetic acid solution. Then, 0.2 mol/L of sodium acetate trihydrate was prepared using the same method. Next, 3.4 mL of the acetic acid solution was accurately placed in a 25 mL volumetric flask, and the sodium acetate solution was added to the mark to obtain an acetic acid–sodium acetate solution with pH 5.5.

### 4.3. Methods for Chromogenic Reaction and Content Determination

Next, 0.5 mL of the test supernatant, 1.5 mL of the 5% aluminum chloride solution, and 1.0 mL of the acetic acid–sodium acetate solution were placed in a 10 mL vial, and 50% ethanol was added to the mark to prepare the reaction solution, which was subsequently reacted for 30 min at room temperature. Then, the absorbance was determined at 400 nm by using a spectrophotometric method, and the yield (mg/g) of flavonoids was calculated using the following equation according to the absorbance of the reaction solution:$$Yield\;of\;flavonoid=\frac{C\times n\times V}{W}$$

Here, W is the sample weight (g) of PR, *n* is the dilution rate, *V* is the sampling volume (mL) of the test supernatant, and C is the concentration (mg/mL) of the test supernatant. All tests were carried out in triplicate.

### 4.4. Method Validation

The relative standard deviation (RSD) of the flavonoid yield from the sample solution was used to analyze the method validation. A sample solution of PR (S8) was accurately prepared using the method in Section 4.2 for method validation, and the absorbance of the standard solution was determined five times to evaluate the precision. Similarly, repeatability was determined by analyzing the sample solution six times, and stability was estimated by storing the sample solution at room temperature and analyzing it at 0, 2, 4, 8, 10, and 12 h. Subsequently, the flavonoid yield of the six sample solutions that contained 3.0 mg of the standard rutin powder was determined, and the sample recovery was investigated.

### 4.5. Determination of Single-Factor Experiment Conditions

The fixed conditions of the initial extraction process optimization were as follows: 50% ethanol, a volume of 25 mL, an extraction temperature of 60 °C, and an ultrasound extraction time of 60 min. Then, each of these factors was investigated with the others remaining unchanged. Therefore, an extraction time of 30–70 min, a solid–liquid ratio of 1:15–1:35 g/mL, and an ethanol concentration of 40–80% were selected as the influencing factors in the single-factor experimental design to study their influence on flavonoid yield [40]. A sample solution of PR (S8) was accurately prepared in accordance with the method in Section 4.2 for the single-factor experiment, and the flavonoid content was determined using the method in Section 4.3.

### 4.6. Response Surface Methodology (RSM) Optimization Test

The preliminary ranges of the extraction time, solid–liquid ratio, and extraction temperature as extraction variables were determined based on the results of the single-factor experiment. In this study, for condition optimization, the principle of BBD was used to test the effect of the independent variables (A, extraction time; B, solid–liquid ratio; C, ethanol concentration), which consisted of three factors and three levels [41]. Then, the flavonoid yield of PR was used as the response value to optimize the extraction process parameters using Design-Expert 13.0 software. The factors and levels are presented in Appendix A.

### 4.7. Determination of Antioxidant Activity

A sample solution was prepared according to the optimal extraction process in Section 4.6, and it was used to determine the antioxidant activity. All tests were carried out in triplicate.

#### 4.7.1. Determination of DPPH Radical-Scavenging Capacity

The DPPH radical-scavenging activity was determined according to Alara R O et al., with slight modifications [42]. Briefly, 1.0 mL of 0.1 mmol/L DPPH free radical solution and 1.0 mL of 50% ethanol were added to a vial to react for 10 min at a room temperature of 25 °C in the dark. The absorbance was determined at 517 nm and recorded as A_0_. Similarly, various volumes of the sample solution and 1.0 mL of the DPPH free radical solution were added to a vial and supplemented with 50% ethanol to 2.0 mL. Then, the absorbance of the sample solution was determined using the same reaction method and recorded as A. The DPPH radical-scavenging activity was calculated using the following equation:$$DPPH\;radical\;scavenging\;rate\;\left(\%\right)=\frac{A_{0}-A}{A_{0}}\times 100\%$$

#### 4.7.2. Determination of ABTS Radical-Scavenging Capacity

The ABTS radical-scavenging activity was evaluated according to a method described by Mohammad N et al. [43]. First, 5.40 mM ABTS stock solution and 1.90 mM K_2_S_2_O_4_ stock solution were taken at equal volumes and reacted at room temperature in a dark environment for 12 h to prepare a working solution of ABTS. The absorbance of the ABTS working solution was adjusted to 0.7 before use. Then, 0.4 mL of 50% ethanol as a blank solvent and 1.6 mL of the ABTS working solution were mixed and reacted for 10 min at 25 °C, and the absorbance was recorded as A_i_. Similarly, different volumes of the sample solution and ABTS working solution were mixed, diluted to 2 mL with 50% ethanol, and reacted for 10 min. The absorbance was determined at 734 nm and recorded as A_ii_. The ABTS free radical-scavenging activity was calculated according to the following formula:$$ABTS\;radical\;scavenging\;rate\;\left(\%\right)=\frac{A_{i}-A_{ii}}{A_{i}}\times 100\%$$

### 4.8. UHPLC Analysis

#### 4.8.1. Chromatographic Analysis Conditions

A UHPLC analysis was performed on an Agilent ultra-high-performance liquid chromatography system (Agilent, Santa Clara, CA, USA) equipped with a binary solvent delivery pump, an autosampler, a diode-array detector (DAD), and a ZORBAX RRHD Eclipse Plus C18 column (2.1 × 50 mm, 1.8 mm, USDAY33948). The mobile phase consisted of eluent A (0.01% formic acid aqueous solution) and eluent B (0.01% formic acid acetonitrile) at a constant flow rate of 0.2 mL/min, with the following elution program: 0–5 min, 10% to 13% solvent B; 5–15 min, 13% to 22% solvent B; 15–20 min, 22% to 30% solvent B; 20–25 min, 30% to 40% solvent B; and 25–30 min, linear gradient from 40% to 99% solvent B. The other chromatographic conditions were as follows: an injection volume of 2 µL, a column temperature of 25 °C, and a detection wavelength of 290 nm.

#### 4.8.2. Preparation of Standard Solution and Sample Solutions

For fingerprinting, chlorogenic acid, neochlorogenic acid, caffeic acid, and quercetin were accurately weighed and dissolved in methanol to prepare standard solutions. The sample solution was extracted using the sample preparation method in Section 4.2, and it was concentrated via rotary evaporation to collect extract powder. Then, the extract powder was dissolved with 50% methanol and transferred into a 2 mL sample vial, and the volume was made up to the mark with 50% methanol. Finally, the sample solutions were filtered with a 0.22 μm membrane filter for a UHPLC analysis.

#### 4.8.3. System Suitability Investigation

The applicability of a number of chromatographic parameters (column, wavelength of detection, analysis time, mobile phase, mobile phase gradient elution, mobile phase flow rate, and column temperature) was investigated, with the number, resolution, and shape of the chromatography peaks and the response value of the target compound selected as conditions in the experimentation.

#### 4.8.4. Verification of Methodology of UHPLC Fingerprinting Analysis

A sample solution of PR was prepared using the method in Section 4.8.2 to verify the fingerprinting methodology, and the relative area of common peaks was selected as the detection metric. Then, the RSD value was calculated to validate the UHPLC fingerprinting method [44]. Firstly, 50% methanol was selected as the blank reference solvent to be injected into the UHPLC system in order to determine the effect of the mobile phase on the sample analysis according to the chromatographic conditions in Section 4.8.1. Thereafter, for a precision analysis, a sample solution (S28) was injected into the chromatographic instrument six times. Similarly, six sample solutions of the same batch (S28) were prepared, and the repeatability was evaluated. At the same time, the sample solution (S28) was injected into the system six times at 2 h intervals, and its stability was assessed.

### 4.9. Chemometrics Analysis

Twenty-eight batches of PR samples were prepared using the method in Section 4.8.2, and chromatographic profiles were obtained under the chromatographic conditions in Section 4.8.1. The AIA format file of the PR samples obtained from the UHPLC analysis system were input into the “Chromatographic Fingerprint Similarity Evaluation System of Traditional Chinese Medicine” (CFSESTCM, version 2012), and then fingerprints were automatically generated. The obtained controlled chromatograms had 35 common peaks, and a similarity computation was performed using multi-point correction and peak matching. Meanwhile, the data of the peak areas from the chromatographic fingerprinting were processed with Z-score standardization in SPSS software (26.0 version), and they were used in an HCA and PCA [45].

### 4.10. Spectrum–Effect Relationship

In this study, the spectrum–effect relationships between the common peaks in the chromatographic fingerprints and the antioxidant activity of PR were examined using a gray relation analysis (GRA), a Pearson correlation analysis, and a partial least-squares discriminant analysis (PLS-DA).

#### 4.10.1. Data Preprocessing

The antioxidant indices were used as the reference series, and the areas of the 35 common characteristic peaks were used as the comparison sequences. The dimension between the values of the sequences to be measured was inconsistent, which may have led to uncredible evaluation results, thereby directly affecting the correctness of conclusions. Thus, the preprocessing and normalization of the original data were performed using the Z-score in SPSS 26.0 software before examining the spectrum–effect relationship in the GRA, Pearson correlation analysis, and PLS-DA.

#### 4.10.2. Gray Relational Grade Calculation

In the GRA, the reference sequences, comparison sequences, and absolute difference sequences were represented by {X_0_ (n)}, {X_i_ (n)}, and Δ_0i_(k) (with i representing the number of samples), respectively. k was designed as an evaluation index sequence (k  =  1, 2, …, n) to obtain the evaluation sequence {X_i_ k} (i = 2; n = 35, in this study). The reference sequences and comparison sequences were recorded as {X_0_ (k)} and {X_i_ (k)} when n = k (with k representing the peak). Furthermore, the gray relational coefficients (η(k)) and gray relational grade (r) for each common peak were calculated at the same time according to the following expression [46]:M = Δmax = max maxΔ_0i_(k)(1)
M = Δmin = min minΔ_0i_(k)(2)
Δ_0i_(k) = ∣X_0_(k) − X_i_(k)∣(3)
η(k) = m + ρM/Δ_0i_(k) + ρM(4)
r_i_ = r_i(0)_/[r_i(0)_ + ri_(k)_](5)

Here, M and n are the optimal and worst values in the absolute difference sequence, and ρ is the resolution coefficient, which is usually set to 0.5. Similarly, the data were processed with the method in Section 4.10.1. Origin2021 software was used to analyze the Pearson correlation analysis results, and Simca14.0 software was used to analyze the PLS-DA results.

### 4.11. Statistical Analysis

In this study, the weighing of the PR samples and the determination of the flavonoid content and antioxidant activity were repeated in parallel in triplicate and averaged, and the data are presented as mean and standard deviations ($\bar{X}$ ± SD). The collection, processing, and analysis of the data and the construction of figures were carried out using SPSS (26.0 version), Simca (14.1 version), Origin (2021 version), and GraphPad (9.5.0 version) software. Additionally, Design Expert 13.0 software was used to design the Box–Behnken response surface trial and analyze the regression equation. *p* < 0.05 was considered statistically significant.

## 5. Conclusions

In this study, the ultrasound-assisted flavonoid extraction process from PR was successfully optimized using Box–Behnken response surface design. The optimum extraction conditions were as follows: an extraction solvent of 53% ethanol, an ultrasound extraction time of 60 min, and a solid–liquid ratio of 1:26 (g/mL). Meanwhile, antioxidant activity experiments indicated that the flavonoid extract from PR had good antioxidant effects. Then, the typical UPLC fingerprinting of twenty-eight batches of PR was carried out, and a total of 35 common peaks were calibrated. A stoichiometric analysis of the spectrum–effect relationship revealed that the chemical components represented by peaks 4, 12, 21, and 24 may be responsible for the antioxidant activity of PR. This suggests that the antioxidant activity of PR is the result of the synergistic effect of multiple components. In addition, a chemometric analysis revealed obvious variations in the content of PR samples of different origins, but there was basic consistency in their chemical composition and antioxidant activity. Not only does this research study provide an effective method for flavonoid extraction from PR, but it also provides a relevant pharmacodynamic theoretical basis for the rational development and utilization of PR as a natural antioxidant. Meanwhile, based on the study of the spectrum–effect relationship, the active components reflecting the antioxidant efficacy of PR were extracted, and chromatographic fingerprints with good stability and chromatographic separation were obtained, providing a scientific reference for the screening of antioxidant quality markers and the quality control standards of PR.

## Figures and Tables

**Figure 1 molecules-29-03860-f001:**
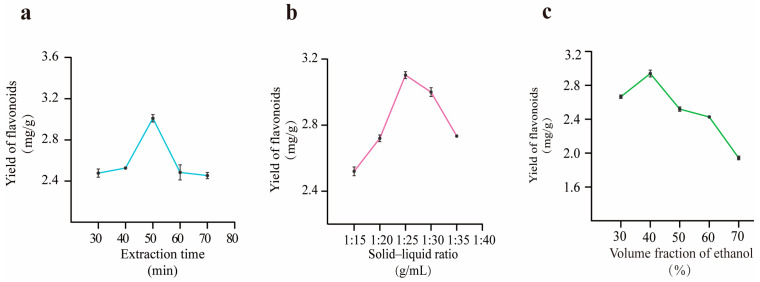
Single-factor test results ((**a**): extraction time; (**b**): solid–liquid ratio; (**c**): volume fraction of ethanol).

**Figure 2 molecules-29-03860-f002:**
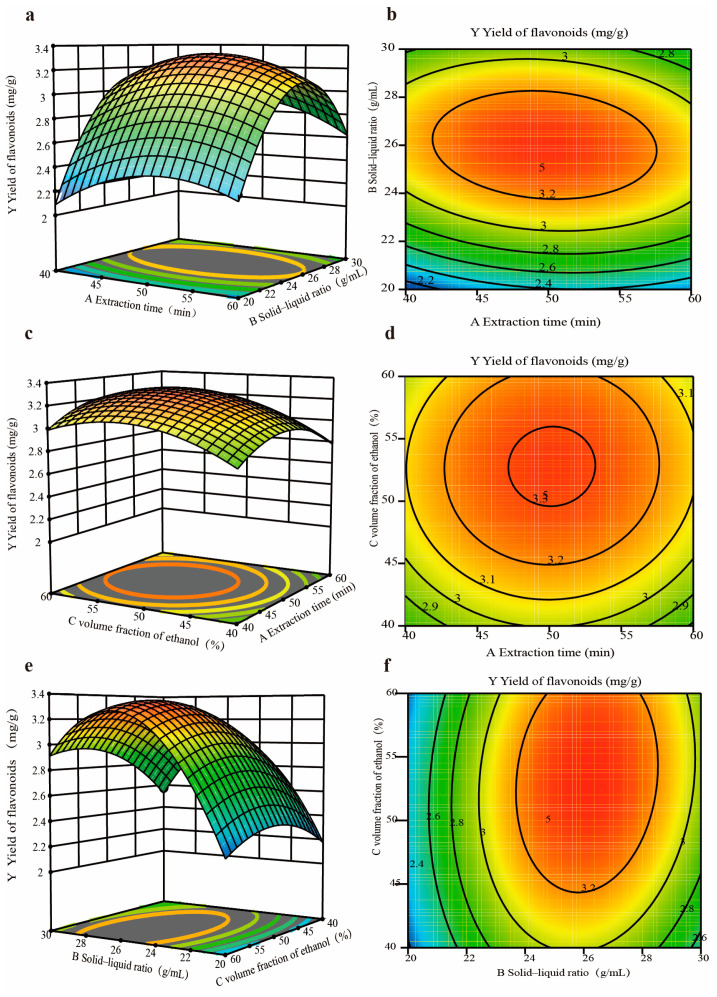
Response surface interaction result plots ((**a**–**f**): the effects of the variables on the yield of flavonoids).

**Figure 3 molecules-29-03860-f003:**
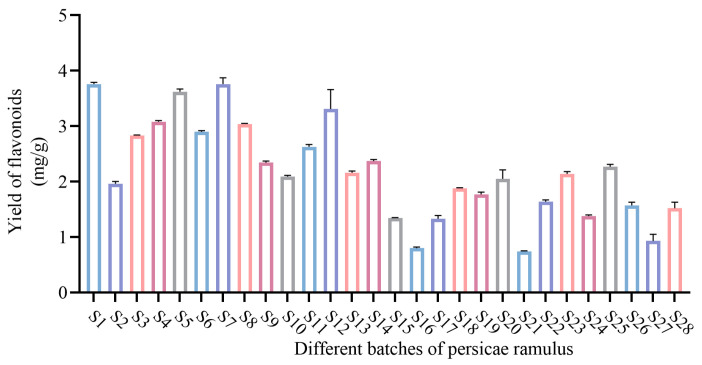
Yield of flavonoids from different batches of persicae ramulus.

**Figure 4 molecules-29-03860-f004:**
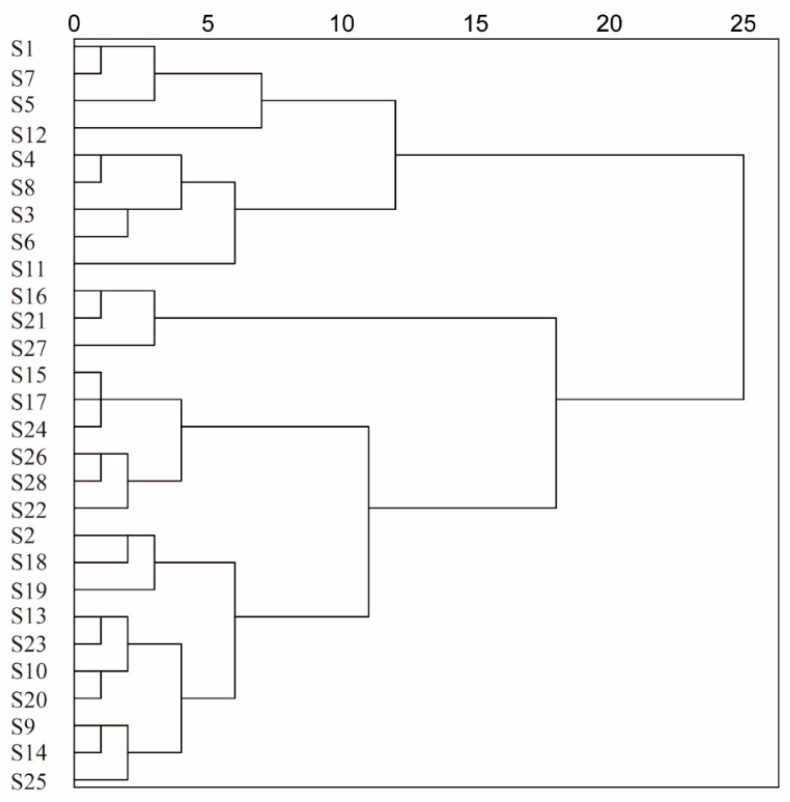
HCA of flavonoid yield from different batches of persicae ramulus.

**Figure 5 molecules-29-03860-f005:**
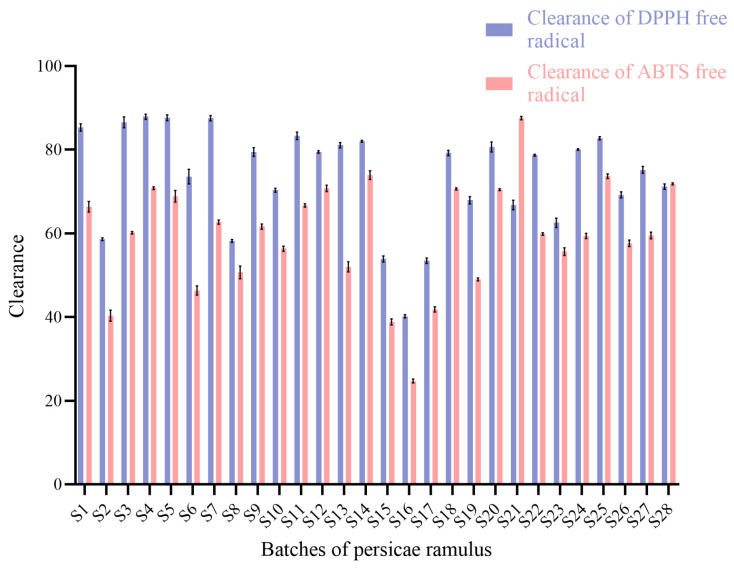
Free radical clearance of flavonoids from persicae ramulus.

**Figure 6 molecules-29-03860-f006:**
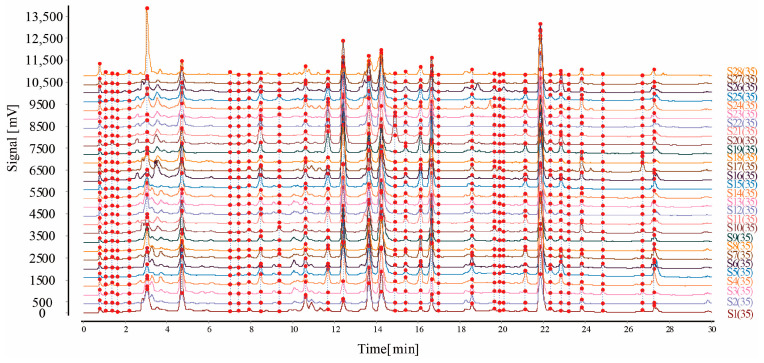
Superimposed chromatograms of twenty-eight batches of persicae ramulus.

**Figure 7 molecules-29-03860-f007:**
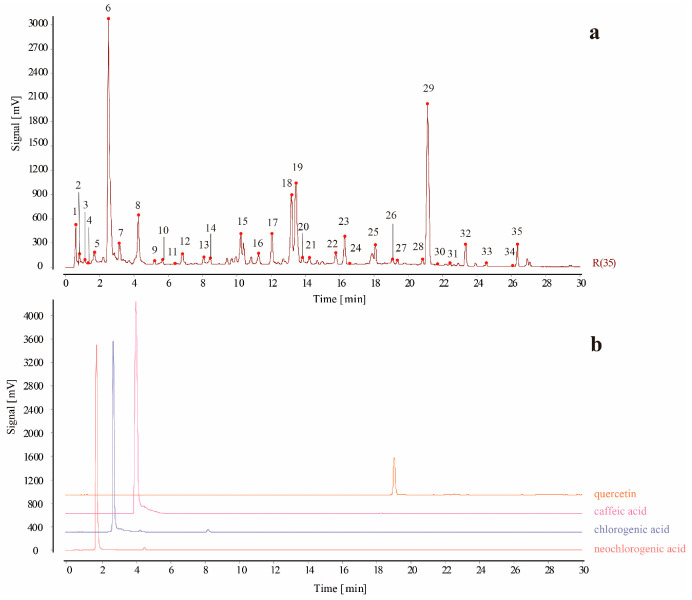
(**a**) UHPLC control fingerprinting of persicae ramulus (the numbers 1–35 represent common peaks in the twenty-eight batches of PR); (**b**) UHPLC fingerprinting of reference solution.

**Figure 8 molecules-29-03860-f008:**
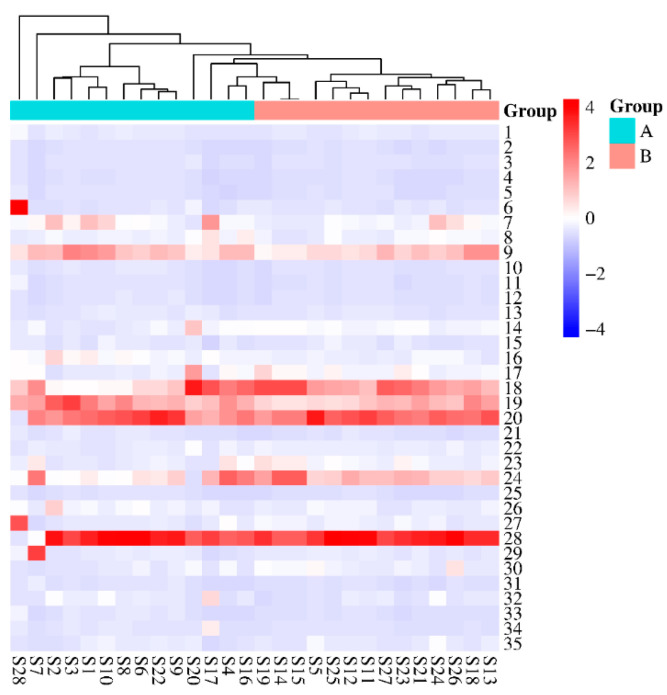
HCA result of UHPLC.

**Figure 9 molecules-29-03860-f009:**
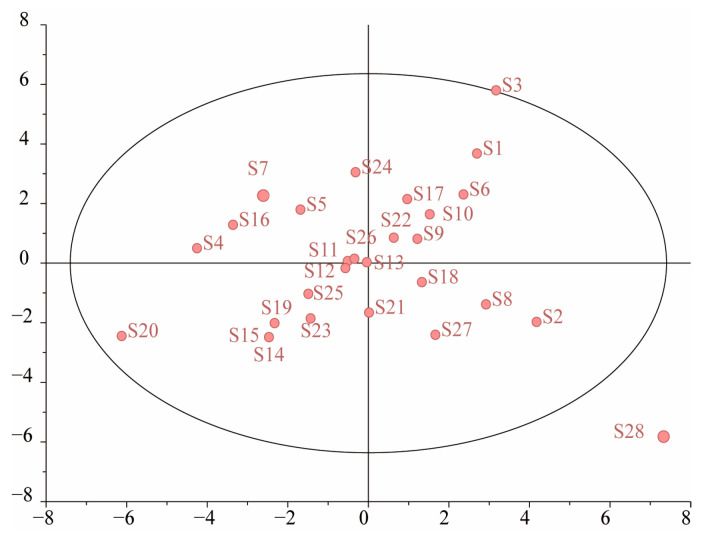
PCA score plot of persicae ramulus samples.

**Figure 10 molecules-29-03860-f010:**
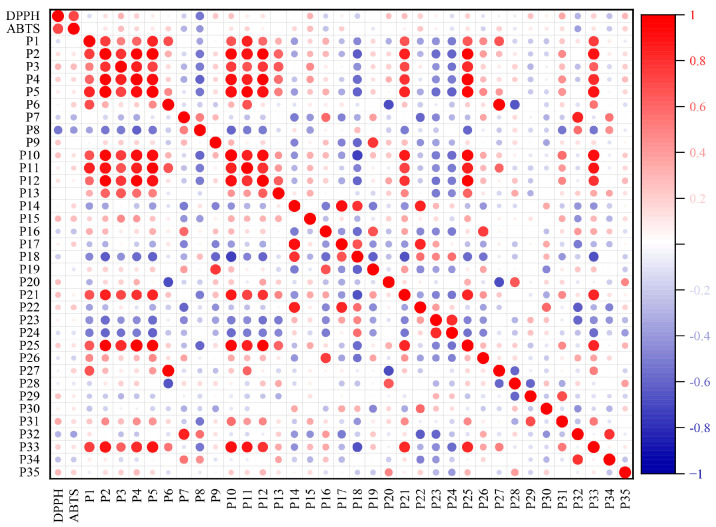
Pearson correlation analysis of 35 common peak areas and antioxidant activities (red indicates a positive correlation and blue indicates a negative correlation).

**Figure 11 molecules-29-03860-f011:**
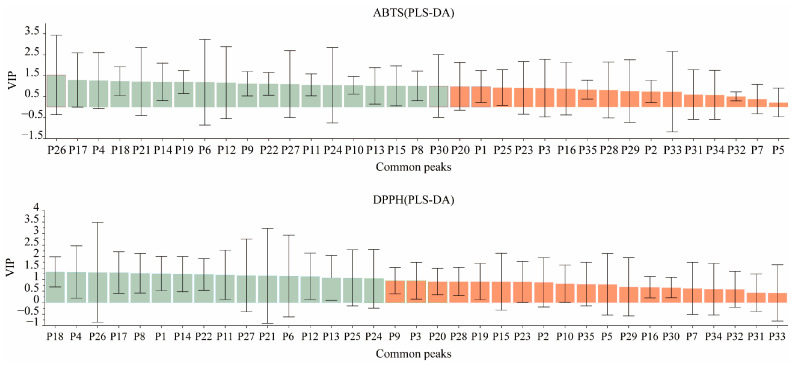
The VIP contribution of 35 common peaks to the antioxidant activity of persicae ramulus (green indicates VIP > 1, and orange indicates VIP < 1).

**Figure 12 molecules-29-03860-f012:**
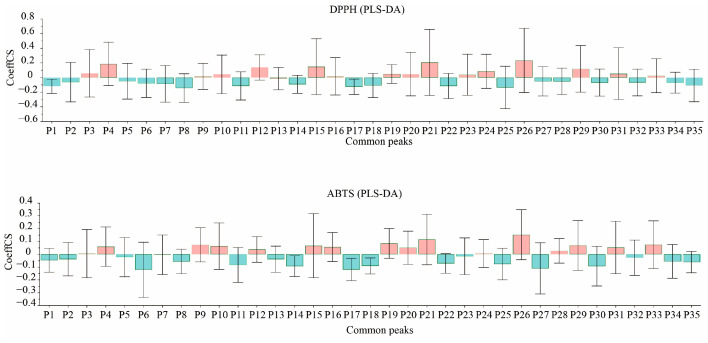
The correlation coefficients of 35 characteristic peaks with antioxidant activity (orange indicates positive standard regression coefficients, and blue indicates a negative standard regression coefficients).

**Table 1 molecules-29-03860-t001:** Similarities between the twenty-eight batches of persicae ramulus.

Samples	Similarities	Samples	Similarities	Samples	Similarities
S1	0.898	S11	0.944	S21	0.949
S2	0.860	S12	0.941	S22	0.924
S3	0.861	S13	0.939	S23	0.937
S4	0.900	S14	0.947	S24	0.939
S5	0.925	S15	0.873	S25	0.941
S6	0.932	S16	0.920	S26	0.943
S7	0.928	S17	0.863	S27	0.936
S8	0.903	S18	0.942	S28	0.948
S9	0.938	S19	0.894		
S10	0.903	S20	0.801		

**Table 2 molecules-29-03860-t002:** Results of gray relational coefficient of 35 common peaks.

Peak	Correlation Coefficient		Peak	Correlation Coefficient	
	DPPH	ABTS		DPPH	ABTS
1	0.97	0.96	19	0.96	0.95
2	0.96	0.95	20	0.96	0.95
3	0.95	0.94	21	0.94	0.93
4	0.94	0.94	22	0.94	0.94
5	0.94	0.93	23	0.92	0.92
6	0.90	0.89	24	0.93	0.93
7	0.92	0.91	25	0.94	0.93
8	0.94	0.94	26	0.95	0.95
9	0.96	0.95	27	0.92	0.92
10	0.96	0.95	28	0.96	0.95
11	0.95	0.95	29	0.92	0.92
12	0.96	0.96	30	0.92	0.91
13	0.96	0.95	31	0.92	0.91
14	0.93	0.93	32	0.92	0.91
15	0.93	0.92	33	0.93	0.92
16	0.95	0.94	34	0.94	0.93
17	0.92	0.92	35	0.93	0.92
18	0.94	0.94			

## Data Availability

The data presented in this study are available in the article and Appendix A.

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
