# Peer review of "Optimization and Spectrum–Effect Analysis of Ultrasonically Extracted Antioxidant Flavonoids from Persicae Ramulus"

_molecules, 2024, doi:10.3390/molecules29163860_

Round 1

Reviewer 1 Report

Comments and Suggestions for Authors The manuscript entitled "Optimization of extraction process and the antioxidant activity spectrum–effect relationship research of flavonoids from persicae ramulus" optimized the ultrasonic-assisted extraction process of flavonoids of PR and established the fingerprinting profile to analyze the spectrum-effect relationship of antioxidant activity. The experiments are well plan, the ideas and methods are scientific rigorous. It will attract the attention of a wide readership. In my mind, the manuscript is acceptable for publication in Molecules after major revision 1. ABSTRACT section need add some important data to support the conclusion.

2. Line 45-46 Among them, flavonoids contained good pharmacological research value and development and utilization potential [4,5,6]. 6 Hujun Xie, Chengzhi Liu, Jian Gao, Jieyu Shi, Fangfang Ni, Xin Luo, Ying He, Gerui Ren, Zisheng Luo. Fabrication of Zein-Lecithin-EGCG complex nanoparticles: Characterization, controlled release in simulated gastrointestinal digestion. Food Chemistry 365 (2021) 130542

3. Line 50-52 Ultrasound-assisted extraction (UAE) was a method that using the acoustic cavitation effect product by ultrasonic to promote the fast dissolution and precipitation of components [6,7,8]. 8 Xizhe Fu, Di Wang, Tarun Belwal, Jing Xie, Yanqun Xu, Li Li, Ligen Zou, Lixia Zhang, Zisheng Luo. Natural deep eutectic solvent enhanced pulse-ultrasonication assisted extraction as a multi-stability protective and efficient green strategy to extract anthocyanin from blueberry pomace. LWT 144 (2021) 111220

4. Line 73 Flavonoids was a class of compounds with C6-C3-C6 structure as basic skeleton [16], 16 Huang, H., Zhu, Y., Fu, X., Zou, Y., Li, Q., & Luo, Z. (2022). Integrated natural deep eutectic solvent and pulse-ultrasonication for efficient extraction of crocins from gardenia fruits (Gardenia jasminoides Ellis) and its bioactivities. Food Chemistry, 380, 132216

5. Figures caption need more illustration.

6. The English language of manuscript should improve.

Comments on the Quality of English Language

need revise by native speak lnglish

Author Response

Dear reviewer

First, thank you very much for your valuable amendments to this article. At present, we have revised the opinions according to you, and answered your question. Furthermore, for English editing, we have revised the manuscript accordingly through the English revision service provided by this journal.

Comments1. ABSTRACT section need add some important data to support the conclusion.

Response1:Thank you very much to the reviewers for the suggested changes,We have revised the summary according to your recommendations.

Comments2. Line 45-46 Among them, flavonoids contained good pharmacological research value and development and utilization potential [4,5,6]. 6 Hujun Xie, Chengzhi Liu, Jian Gao, Jieyu Shi, Fangfang Ni, Xin Luo, Ying He, Gerui Ren, Zisheng Luo. Fabrication of Zein-Lecithin-EGCG complex nanoparticles: Characterization, controlled release in simulated gastrointestinal digestion. Food Chemistry 365 (2021) 130542

Response2: Thanks to the reviewers for your comments,We have revised and cited based on the literature you provide in line 45-46.

Comments3. Line 50-52 Ultrasound-assisted extraction (UAE) was a method that using the acoustic cavitation effect product by ultrasonic to promote the fast dissolution and precipitation of components [6,7,8]. 8 Xizhe Fu, Di Wang, Tarun Belwal, Jing Xie, Yanqun Xu, Li Li, Ligen Zou, Lixia Zhang, Zisheng Luo. Natural deep eutectic solvent enhanced pulse-ultrasonication assisted extraction as a multi-stability protective and efficient green strategy to extract anthocyanin from blueberry pomace. LWT 144 (2021) 111220

Response3: Thanks to the reviewers for your comments,We have revised and cited based on the literature you provide in line 50-52.

Comments4. Line 73 Flavonoids was a class of compounds with C6-C3-C6 structure as basic skeleton [16], 16 Huang, H., Zhu, Y., Fu, X., Zou, Y., Li, Q., & Luo, Z. (2022). Integrated natural deep eutectic solvent and pulse-ultrasonication for efficient extraction of crocins from gardenia fruits (Gardenia jasminoides Ellis) and its bioactivities. Food Chemistry, 380, 132216

Response4: Thanks to the reviewers for your comments,We have revised and cited based on the literature you provide in line 73.

Comments5. Figures caption need more illustration.

Response5: Thanks to the reviewers for your comments,Following your suggestion, we have revised the title and notes for the chart.

Comments6. The English language of manuscript should improve.

Response6: Thank you for your revision to this article and we have revised it through the English editorial service provided in this article.

Reviewer 2 Report

Comments and Suggestions for Authors

There are some issues that should be carefully addressed by authors before making the paper suitable for publication in the Molecules.

Abstract is too long (e.g. peak number are not important for the summary).

English language should be improved (Line 39: is the dried branch NOT was the dried branch; Line 43: PR contains the constituents NOT PR contained the constituents, etc.).

Line 47: The references are missing.

Line 54: Please define ‘’inappropriate ultrasonic extract condition’’.

Line 55: Please use the full name and RSM in the parenthesis when mentioning for the first time. The same comment for DPPH, ABTS….

Line 99: UPLC is trademark from Waters, please use UHPLC.

Lines 106-112: I suggest to delete.

Please do not repeat the same data from Figures in Tables.

Line 250: Concentration of what?

Line 267: The main concern is lack of mass spectrometry for peak identification since PR is complex matrix.

Line 293: What was the control solution?

In Discussion section it might be useful to compare/discuss the results with other authors.

Lines 512-518: Please delete.

Lines 223 and 527: Twenty-eight NOT 28.

Line 532: How did you prepare the powder of PR?

In Tables, please use the full name for PR.

Table 1: Unit for yield is missing. Precision, Stability, Repeatability, and Recovery calculation should be explained in footnote.

Tables 2, 3: Please define A, B, C.

Table 4: Unit for yield is missing.

Table 5: Unit for DPPH and ABTS is missing.

Author Response

Dear reviewer

First, thank you very much for your valuable amendments to this article. At present, we have revised the opinions according to you, and answered your question. Furthermore, for English editing, we have revised the manuscript accordingly through the English revision service provided by this journal.

Comments1:Abstract is too long (e.g. peak number are not important for the summary).

Response1:Thanks to the teacher for making better suggestions for the problems existing in the abstract of this article. Regarding the long summary, we have made appropriate revisions in this article.

Comments2:English language should be improved (Line 39: is the dried branch NOT was the dried branch; Line 43: PR contains the constituents NOT PR contained the constituents, etc.).

Response2:Thanks to reviewer for her comments on the inappropriate language use of the article. Regarding the English level of this article, we have solved the problem through the English editor provided by this magazine and made modifications in the article.

Comments3:Line 47: The references are missing.

Response3: Thanks to reviewer for finding a missing reference, we have added a reference of [3] In the corresponding position to the article.

Comments4:Line 54: Please define ‘’inappropriate ultrasonic extract condition’’.

Response4: Thanks to reviewer for finding express inappropriate words. In the article, we want to express that the inappropriate ultrasound extraction conditions have an impact on the yield of the target extract. In the writing process, due to the English proficiency problem, it was expressed as inappropriate ultrasonic extract condition, it has been modified to the inappropriate ultrasound extraction conditions in line 54 of the article.

Comments5:Line 55: Please use the full name and RSM in the parenthesis when mentioning for the first time. The same comment for DPPH, ABTS….?

Response5:Thanks to the reviewer for the suggestion that the first word should be written in the full name. We have revised it in the introduction section of the original article.

Comments6:Line 99: UPLC is trademark from Waters, please use UHPLC.

Response6: Thanks to the reviewer for your suggestions, we have changed the expression of UPLC in the article to UHPLC.

Comments7:Lines 106-112: I suggest to delete.

Please do not repeat the same data from Figures in Tables.

Response7: Thanks to the reviewers for your suggestions, we have deleted lines 106-112 from the original article.

Comments8:Line 250: Concentration of what?

Response8:Thanks to the reviewers for your question, We have added the varying concentrations (0.03-0.15 mg/mL) of PR flavonoids extracts in 250 lines in the original article.

Comments9:Line 267: The main concern is lack of mass spectrometry for peak identification since PR is complex matrix.

Response9: Thanks to the question of the reviewer, some of the peaks have been identified using standard controls, and the mechanism of antioxidant effect will be further investigated.

Comments10:Line 293: What was the control solution?

Response10: Thanks to the reviewer for your question. The control solution refers to the four standard solutions of new chlorogenic acid, chlorogenic acid, caffeic acid and quercetin in the fingerprinting.

Comments11:In Discussion section it might be useful to compare/discuss the results with other authors.

Response11: Thanks to the reviewers for your suggestions, we have revised the discussion section of the article based on your recommendations by referring to the recommendations of other reviewers.

Comments12:Lines 512-518: Please delete.

Response12: Thanks to the reviewers for your suggestions, we have deleted lines 512-518 from the original article.

Comments13:Lines 223 and 527: Twenty-eight NOT 28.

Response13: Thanks to the reviewers for your suggestions, we have revised the expression of 28 in the article to Twenty-eight.

Comments14:Line 532: How did you prepare the powder of PR?

In Tables, please use the full name for PR.

Response14: Thanks to the reviewers for your questions, PR samples were dried and crushed into powder, and passed through a 65 meshs drug sieve,and we have revised the expression of PR in the tables to persicae ramulus.

Comments15:Table 1: Unit for yield is missing. Precision, Stability, Repeatability, and Recovery calculation should be explained in footnote.

Tables 2, 3: Please define A, B, C.

Table 4: Unit for yield is missing.

Table 5: Unit for DPPH and ABTS is missing.

Response15: Thanks to the reviewers for your questions, we have defined A, B, C in tables 2, 3, and added the unit for yield in table 4, and added the unit for DPPH and ABTS in table 5.

Reviewer 3 Report

Comments and Suggestions for Authors

Generally an interesting paper with well-written English, recommend publishing after major revisions

Please see the attached files of comments.

Comments on the Quality of English Language

Minor editing of English language required

Author Response

Dear reviewer

First, thank you very much for your valuable amendments to this article. At present, we have revised the opinions according to you, and answered your question. Furthermore, for English editing, we have revised the manuscript accordingly through the English revision service provided by this journal.

Comments1: Abstract

The abstract should be rewritten. The abstract should be a clear, concise (150~200

words), one-paragraph summary, informative rather than descriptive, giving scope and

purpose, experimental approach, significant results, and major conclusions. At present, it is too long and unfocused.

Response1: Thanks to the reviewers for your revision comments and questions, and we have appropriately revised the abstract based on your revision comments.

Comments2:Results

Page 6 lines 260: Please confirm that this conclusion, “the clearance was positively

correlated with the flavonoids content in the extracts”, is correct. In Figure 3 and

Table S4, yield of flavonoids in S16 (0.80 mg/g) and S21 (0.74 mg/g) are similar, but

the antioxidant activities of them are very different. In particular, S21 had the lowest

flavonoids extraction rate but the highest rate of ABTS free radical scavenging.

Page 9, Figure 7: Based on the retention time of the standard, please confirm

whether peak 26 is quercetin. Confirmation of the result by LCMS is recommended if

possible.

Response2:Thanks to the reviewers for finding the conclusions contained in the article " the clearance was positively correlated with the flavonoids content in the extracts " Error problem, through your feedback, we have carefully checked this conclusion and modify it. In Figures 3 and Table 4, S16 and S21 have similar content, but they have differences in free radical clearance, and S21 is the strongest for ABTS radical clearance, because there are differences between the reaction principles of each radical and some limitations. Therefore, different results may be obtained even by using the same extraction solution for different radical clearance, so the conclusion has been changed to change that peach flavone extract has different degrees of correlation with DPPH and ABTS. Therefore, S21 has a low yield, but a strong clearance of ABTS radical. Thanks to the reviewer for the question, we repeated the standard control test and identified peak 26 as quercetin. On page 9, in the control fingerprinting of Figure 7, there is some difference between the retention time of quercetin and peak 26, because the column used was blocked for a long time, causing the inconsistency of peak peak time of quercetin standard and 26 in the control atlas. After submission, we have taken steps to maintain the column and again identified peak 26 as quercetin.

Comments3:Materials and Methods

Authors need to pay attention to the correct use of English, such as not using Arabic

numerals at the beginning of sentences.

Response3: Thanks to the reviewer for raising the problem of using Arabic numerals at the beginning of the sentence in this article, we have modified the errors in the corresponding positions and converted them to English.

Comments4:Figures and Tables

More explanation needs to be added into Figure caption as it is currently too

concise. Same for Tables, including supplementary materials.

Many Tables lack a lot of information, e.g. units, explanation of abbreviations,

number of repetitions of parallel trials, etc.

Response4:Thanks to the reviewer for raising the problem of using Arabic numerals at the beginning of the sentence in this article, we have modified the errors in the corresponding positions and converted them to English.

Comments5: References

Many references with formatting errors and incomplete information, should be

corrected.

Response5: Thanks to the reviewer's modifications, we have made changes in the references based on the reviewer's comments and the requirements of the reference citation format of this journal.

Reviewer 4 Report

Comments and Suggestions for Authors

The manuscript entitled "Optimization of Extraction Process and Antioxidant Activity Spectrum-Effect Relationship Research of Flavonoids from Persicae Ramulus" presents a comprehensive study on optimizing the ultrasonic-assisted extraction process of flavonoids from Persicae ramulus and analyzing their antioxidant activities using advanced chromatographic and chemometric techniques. The study is well-designed and employs appropriate methodologies, including Response Surface Methodology (RSM) and Ultra Performance Liquid Chromatography (UPLC), to achieve its objectives. However, there are some improvement opportunities, here. I present my comments:

The title "Optimization of extraction process and the antioxidant activity spectrum–effect relationship research of flavonoids from Persicae ramulus" is informative and accurately reflects the content of the study. Anyway, it could be slightly refined for clarity, as follows: "Optimization and Spectrum-Effect Analysis of Ultrasonically Extracted Antioxidant Flavonoids from Persicae Ramulus".

The introduction section does not clearly articulate the specific research gap that this study aims to fill. It mentions that the optimization of extraction techniques for flavonoids is lacking but does not sufficiently explain why this is important or how it impacts the field. Clearly stating the gap in the literature and how this study addresses it would strengthen the rationale for the research. The novelty of the study should be highlighted more explicitly. What makes this study different from previous research? What new insights or advancements does it offer?

The objectives are mentioned but not explicitly stated as research questions or hypotheses. A clear statement of the research hypothesis or specific aims would provide a better framework for the study.The introduction should clearly state the specific objectives of the study in a concise manner.

The introduction has several grammatical errors and typographical mistakes that need correction.

In M&M section, the ultrasonic-assisted extraction process is described, but key details are missing. For example, the power and frequency of the ultrasound device used should be specified. The methodology should include control experiments to validate the extraction process. Describing these controls would add robustness to the study. The procedures for DPPH and ABTS assays are described but lack specific details. For example, the concentration of the reagents, the incubation times, and the temperature at which the assays were conducted should be specified. Information about the calibration curves and standards used for these assays is missing. Including this information would ensure the assays' reliability and reproducibility. The use of chemometric techniques such as HCA, PCA, and PLS-DA is mentioned, but these methods are not adequately explained. Briefly describing each technique and its purpose in the context of the study would be beneficial. The number of replicates for each experiment should be clearly stated. This information is crucial for assessing the reliability of the results. A table summarizing the optimized conditions for the extraction process, UPLC analysis, and antioxidant assays would provide a clear overview of the experimental setup.

The optimization results are presented, but the section lacks clarity. Summarize key findings in a table that lists the optimal conditions for each parameter (e.g., extraction time, solvent concentration, solid-liquid ratio). The section should include a detailed statistical analysis of the optimization results. The flavonoid content results are presented, but there is inconsistency in the data presentation. Ensure all data are reported uniformly, preferably in a table format with mean values and standard deviations. Compare the flavonoid content across different batches with statistical tests to highlight significant differences. While the antioxidant activity results are presented, the correlation between flavonoid content and antioxidant activity is not adequately explored. Include statistical correlation analyses (e.g., Pearson correlation coefficient) to establish this relationship. Provide a more detailed interpretation of the chemometric analysis results. Explain the significance of the identified clusters and principal components in relation to the antioxidant activity and flavonoid content.

The discussion section provides a summary of the results but lacks depth in interpreting the data. It should explore the implications of the findings in greater detail, explaining how the optimized extraction process improves upon previous methods and why the identified flavonoid peaks are significant. Integrate more recent studies and review articles to provide a broader context for the findings. Discuss how this study contributes to the current knowledge and fills existing research gaps. Discuss the variability in flavonoid content and antioxidant activity across different PR batches in more detail. Explain how this variability might impact the reproducibility and generalizability of the results. Some parts of the discussion could be more concise. Focus on the key points and avoid unnecessary repetition.

The conclusion is overly general and does not provide specific details about the key findings of the study. It should clearly state the most important results, such as the optimal extraction conditions and the identified antioxidant-active flavonoids. The conclusion fails to emphasize the significance of the findings. Highlight why the optimized extraction process and the identified spectrum-effect relationships are important for the field.

I strongly recommend extensive editing by a native English speaker or a professional editor with expertise in scientific writing. Finally, I recommend that the authors significantly revise the manuscript before it can be reconsidered for publication.

Comments on the Quality of English Language

I recommend that the authors significantly revise the manuscript before it can be reconsidered for publication.

Author Response

Dear reviewer

First, thank you very much for your valuable amendments to this article. At present, we have revised the opinions according to you, and answered your question. Furthermore, for English editing, we have revised the manuscript accordingly through the English revision service provided by this journal.

Comments1:The title "Optimization of extraction process and the antioxidant activity spectrum–effect relationship research of flavonoids from Persicae ramulus" is informative and accurately reflects the content of the study. Anyway, it could be slightly refined for clarity, as follows: "Optimization and Spectrum-Effect Analysis of Ultrasonically Extracted Antioxidant Flavonoids from Persicae Ramulus".

Response1:Thank you very much to the reviewers for the suggested changes, we have changed the title to "Optimization and Spectrum-Effect Analysis of Ultrasonically Extracted Antioxidant Flavonoids from Persicae Ramulus".

Comments2:The introduction section does not clearly articulate the specific research gap that this study aims to fill. It mentions that the optimization of extraction techniques for flavonoids is lacking but does not sufficiently explain why this is important or how it impacts the field. Clearly stating the gap in the literature and how this study addresses it would strengthen the rationale for the research. The novelty of the study should be highlighted more explicitly. What makes this study different from previous research? What new insights or advancements does it offer?

The objectives are mentioned but not explicitly stated as research questions or hypotheses. A clear statement of the research hypothesis or specific aims would provide a better framework for the study. The introduction should clearly state the specific objectives of the study in a concise manner.

Response2:We thank the reviewers for their comments on the introductory part of this article, and we have revised the corresponding position in the article according to your suggestions, and added The aim of this study is to optimize the extraction process of peach twig flavonoids by response surface methodology, and to investigate the spectroeffective relationship of antioxidant of peach twig flavonoids by using the GRA, PLS-DA, and Pearson's correlation of the chemometric analysis methods. The study was conducted to fill the gap in the pharmacological activity of persicae ramulus flavonoid extracts.

Comments3:The introduction has several grammatical errors and typographical mistakes that need correction.

Response3: Thanks to the reviewers for their comments on the introduction section of this article which we have already appropriately revised.

Comments4:In M&M section, the ultrasonic-assisted extraction process is described, but key details are missing. For example, the power and frequency of the ultrasound device used should be specified. The methodology should include control experiments to validate the extraction process. Describing these controls would add robustness to the study. The procedures for DPPH and ABTS assays are described but lack specific details. For example, the concentration of the reagents, the incubation times, and the temperature at which the assays were conducted should be specified. Information about the calibration curves and standards used for these assays is missing. Including this information would ensure the assays' reliability and reproducibility. The use of chemometric techniques such as HCA, PCA, and PLS-DA is mentioned, but these methods are not adequately explained. Briefly describing each technique and its purpose in the context of the study would be beneficial. The number of replicates for each experiment should be clearly stated. This information is crucial for assessing the reliability of the results. A table summarizing the optimized conditions for the extraction process, UPLC analysis, and antioxidant assays would provide a clear overview of the experimental setup.

Response4:Thanks to the reviewers for their comments on the method and materials section of this article, we added the power of 60% and the frequency of 40Hz. For the concentration of Shi Jie, the incubation time, and the temperature measured in the DPPH and ABTS tests, we have added some contents to the corresponding positions in the antioxidant part of this article. In addition, following your suggestion, we have added the introduction of the relevant stoichiometric analysis, the description of the number of repetitions in the trial and the results of the optimization conditions for the extraction optimization process.

Comments5:The optimization results are presented, but the section lacks clarity. Summarize key findings in a table that lists the optimal conditions for each parameter (e.g., extraction time, solvent concentration, solid-liquid ratio). The section should include a detailed statistical analysis of the optimization results. The flavonoid content results are presented, but there is inconsistency in the data presentation. Ensure all data are reported uniformly, preferably in a table format with mean values and standard deviations. Compare the flavonoid content across different batches with statistical tests to highlight significant differences. While the antioxidant activity results are presented, the correlation between flavonoid content and antioxidant activity is not adequately explored. Include statistical correlation analyses (e.g., Pearson correlation coefficient) to establish this relationship. Provide a more detailed interpretation of the chemometric analysis results. Explain the significance of the identified clusters and principal components in relation to the antioxidant activity and flavonoid content.

Response5:Thanks to the reviewers for their revision comments, we have added the optimization results table for each parameter to the corresponding position of the article according to your revision comments. And the flavonoid data were changed to mean values and standard deviations. Furthermore, with your amendments, we added Pearson correlation analysis to the spectrum-effect analysis of this article.

Comments6:The discussion section provides a summary of the results but lacks depth in interpreting the data. It should explore the implications of the findings in greater detail, explaining how the optimized extraction process improves upon previous methods and why the identified flavonoid peaks are significant. Integrate more recent studies and review articles to provide a broader context for the findings. Discuss how this study contributes to the current knowledge and fills existing research gaps. Discuss the variability in flavonoid content and antioxidant activity across different PR batches in more detail. Explain how this variability might impact the reproducibility and generalizability of the results. Some parts of the discussion could be more concise. Focus on the key points and avoid unnecessary repetition.

Response6: Thank you to the reviewer for the discussion of modified comments, we added to the discussion of relevant content according to your suggestion, discussed through the optimization of peach flavonoids extraction process, fill the peach related pharmacology, and the current research on peach only in the beginning, flavonoids as its active ingredients, contains a variety of biological activities. However, there are few studies on the flavonoids of persicae ramulus, which limits the development of this medicine. Therefore, in this paper, by optimizing the extraction process of the flavonoids and studying the antioxidant spectrum relationship, further proves that the flavonoids of persicae ramulus have good antioxidant activity and have significant development and utilization value.

Comments7:The conclusion is overly general and does not provide specific details about the key findings of the study. It should clearly state the most important results, such as the optimal extraction conditions and the identified antioxidant-active flavonoid. The conclusion fails to emphasize the significance of the findings. Highlight why the optimized extraction process and the identified spectrum-effect relationships are important for the field.

Response7: Thanks to the reviewers for their revised suggestions, we have added the optimized optimal extraction conditions and the importance of optimizing the persicae ramulus flavonoid component extraction process in our conclusion.

The conclusion is as following:In this study, the ultrasound-assisted extraction process of flavonoid from PR was successfully optimized using BoxBehnken response surface design. The optimum ex-traction conditions were as follows: an extraction solvent of 53% ethanol, an ultra-sound extraction time of 60 min, and a solid-liquid ratio of 1:26 (g/mL). Meanwhile, antioxidant activity experiments indicated that the flavonoid extract from PR had good antioxidant effects. Then, the typical UPLC fingerprinting of twenty-eight batch-es of PR was carried out, and a total of 35 common peaks were calibrated. A stoichiometric analysis of the spectrum-effect relationship revealed that the chemical components represented by peaks 4, 12, 21, and 24 may be responsible for the antioxidant activity of PR. This suggests that the antioxidant activity of PR is the result of the synergistic effect of multiple components. In addition, a chemometric analysis revealed obvious variations in the content of PR samples of different origins, but there was a basic consistency in their chemical composition and antioxidant activity. Not only does this research provide an effective method for flavonoid extraction from PR, but it also provides a relevant pharmacodynamic theoretical basis for the rational development and utilization of PR as a natural antioxidant. Meanwhile, based on the study of the spec-trum-effect relationship, the active components reflecting the antioxidant efficacy of PR were extracted, and chromatographic fingerprints were obtained with a good stability and chromatographic separation, providing a scientific reference for the screening of antioxidant quality markers and the quality control standards of PR.

Round 2

Reviewer 3 Report

Comments and Suggestions for Authors

No further comments, it is recommended that this version be accepted for publication.